# Simple indices of infarct size post ST-Elevation Myocardial Infarction (STEMI) provides similar risk stratification to cardiac MRI

Lokesh Sharma[1,2], Amir Faour[1,2], Tuan Nguyen[1,2], Hany Dimitri[1,2], Giau Vo[2], James Otton[1,2], Sonya Burgess[1,2,3], Craig Juergens[1,2], John French[1,2,4]*

1 Department of Cardiology, Liverpool Hospital, Liverpool, NSW, Australia, 2 South Western Sydney Clinical School, The University of New South Wales, Sydney, NSW, Australia, 3 Nepean Hospital, Sydney, NSW, Australia, 4 Western Sydney University, Sydney, NSW, Australia

* j.french@unsw.edu.au

**Data Availability Statement:** Deidentified data has been attached as supporting information as Excel.

**Funding:** The author(s) received no specific funding for this work.

## Abstract

### Introduction

Myocardial Infarct Size (IS) determined soon after ST-segment elevation myocardial infarction (STEMI) has prognostic significance, and can be assessed by cardiac biomarker levels, electrocardiographic (ECG) parameters, and imaging modalities (including echocardiography and cardiac magnetic resonance imaging [CMRI]).

### Objectives and methods

We evaluated methods of IS assessment, 12-lead ECG Selvester QRS scores and high-sensitivity Troponin T (hsTnT) levels measured $\geq$48hr (plateau phase of hsTnT elevation), compared to paired CMRIs and echocardiograms, in a prospective cohort of patients with STEMI undergoing percutaneous coronary intervention (PCI) during the index hospitalisation. Associations were determined between IS, as assessed by these methods, and 24-month major adverse cardiac events (MACE), a hierarchical composite of: death, stroke and hospitalization for heart failure.

### Results

Of 233 patients undergoing early CMRI after STEMI, 211 patients (86% male; 54% anterior MI) had first STEMIs, median age 56 years [interquartile range 50–64], of whom 165 (78%) underwent primary PCI and 46 (22%) pharmaco-invasive PCI. Ejection fraction improved from 48% [42–54] acutely to 52% [44–60] at 2 months (p< 0.05). Plateau phase hsTnT levels, QRS scoring and CMRI-determined IS post-STEMI correlated for anterior MIs (all comparisons r>0.4, p<0.01); highest tertiles of these 3 parameters predicted 24 month MACE (log-rank <0.01). Multi-variable binary logistic regression analysis showed 72h hsTnT levels predicted 24-month MACE (p<0.01).

**Competing interests:** The authors have declared that no competing interests exist.

## Conclusion

Post-PCI treatment of STEMI, hsTnT levels measured ≥48h and Selvester QRS scoring correlated with CMRI-determined IS. These parameters predicted MACE at 24 months and should be routinely assessed for post-STEMI risk stratification.

## Introduction

Myocardial Infarct Size (IS) is an important prognostic marker in patients suffering ST-Elevation Myocardial Infarction (STEMI) [1]. For IS assessment in the current era, cardiac magnetic resonance imaging (CMRI) is the 'gold standard'[2–5], though its accessibility may be limited especially when a large proportion of patients undergoing primary PCI or pharmaco-invasive PCI are only in hospital often 2 or 3 days. The Selvester QRS score, developed ~40 years ago, estimates IS from the 12-lead electrocardiogram (ECG), with each point corresponding with 3% of the left ventricular (LV) myocardium affected by myocardial infarction (MI) [6]. This score was initially derived from computer simulation of the human heart activation sequence and validated anatomically in post-mortem studies [7–10]. A strong correlation between IS determined by single-photon emission computed tomography and Selvester QRS score at 7 days post-MI was reported in the thrombolytic era [11].

High sensitivity troponin-T (hsTnT) levels, which have a biphasic release pattern after STEMI correlate with microvascular obstruction (MVO), and IS measured by CMRI [11]. Moreover, plateau-phase hsTnT levels were found to be independent predictors of major adverse cardiovascular events (MACE) and have prognostic utility [2]. In small (<70) patient studies [11–17] correlations between QRS scoring and biomarker-estimated IS were not found.

Thus, we compared Selvester QRS scoring, plateau-phase hsTnT levels and CMRI in first-time STEMI patients and evaluated their associations with late outcomes.

## Methods

### Study population

Patients were screened between May 2012 and August 2015, if they had STEMI treated by primary PCI or pharmaco-invasive PCI at Liverpool Hospital, Sydney, Australia during their initial hospitalization. Patients were eligible if they presented with their first MI, and underwent early and follow-up CMRI, had available and interpretable ECGs at baseline, within 7 days post-infarction and at follow-up, and had 48- and/or 72-hour hsTnT levels (14ng/L upper reference limit [URL] Roche, Basel, Switzerland). Details of the patient population have been previously reported (see consort diagram Fig 1) [18]. STEMI was defined in accordance with the "Fourth universal definition of MI" [19]. Pharmaco-invasive strategy was defined as *fibrinolytic administration followed by rescue PCI or by scheduled angiography and PCI* [20].

The study was approved by the Human Research Ethics Committee at Concord Hospital, Sydney Australia (HREC/11/CRGH/224; approval CH62/6/2011-151). Informed written consent was obtained from each patient, and the study protocol conforms to ethical guidelines from the 1975 Declaration of Helsinki.

### Electrocardiographic analyses

Infarct location was determined from admission 12-lead ECGs with the greatest total sum of ST-elevation with leads V1-6 I and aVL used for anterior MI, leads II, III aVF, V5-6 for inferior

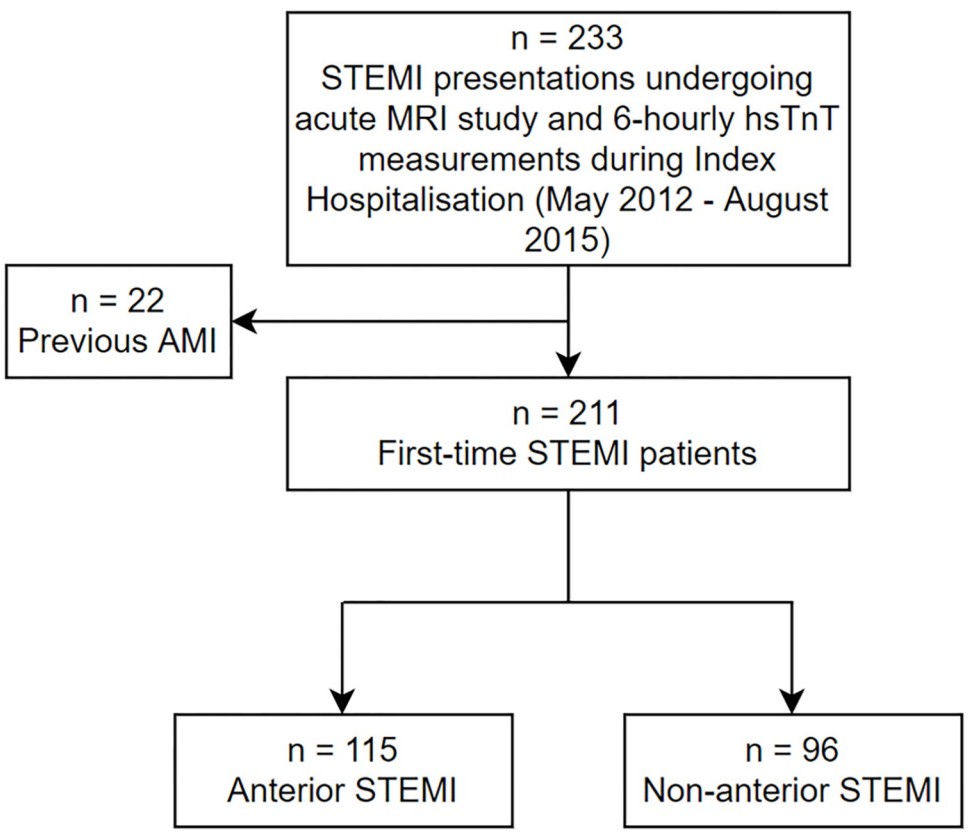

**Fig 1. Consort diagram.** Of the 211 patients with initial STEMI, 9 were missing follow-up ECGs and 9 were missing follow-up MRIs.

and inferoposterior MI and leads V1-3 for posterior MI. Relevant leads showing ≥100-μV ST-segment elevation (or ≥100-μV ST depression in leads V1-3 for inferoposterior infarcts) were assigned the corresponding maximum possible Selvester QRS score and potential myocardium at risk was calculated [9, 10]. Using the same lead positions as the original recording, 12-lead ECGs were recommended 60–90 min post-PCI, 4-8h hours post-reperfusion [21] and at early and follow-up MRI studies. These ECGs were scored by a single observer blinded to CMRI study and clinical data, using the Selvester QRS score as described by Loring et al. with all conduction abnormality criteria, duration and amplitude adjustments being adhered to. In brief, the Selvester QRS score is a 33-point score that estimates IS based on 38 parameters of the QRS waveform such as amplitude, duration and ratios. Each point correlates to 3% of infarcted LV myocardium [22]. Right atrial overload criteria and exclusions were ignored to increase sensitivity of posterior scar size. A cross read of 10% of patients scored was conducted by CPJ.

Qualifying baseline ECGs were ≤120 minutes after reperfusion therapies. The TP segment was taken as the isoelectric line with ST deviation measured to the nearest 0.5- mm in all leads at 80 ms after the J point. Where possible, voltage measurements for each lead were taken as the average ST deviation of 3 complete beats. The voltage in the lead with maximum ST elevation and sum of voltages in all leads with ST deviation on the baseline ECG were recorded. For patients treated with primary PCI, ST recovery measurements were made on an ECG taken 30–90 minutes following the re-achievement of TIMI-3 flow after first successful device use. ST Recovery measurements were undertaken in a) all leads with ≥1 mm ST deviation [23] and b) in the single lead with maximum ST deviation, [24, 25] by JKF(blinded to angiographic

findings and clinical outcomes) with a 10% cross read by AKF; discrepancies were resolved by agreement if necessary with a 3rd reader. TIMI flow grades were assessed by SNB and JKF blinded to ECG findings and clinical outcomes.

## Transthoracic echocardiography acquisition

Transthoracic echocardiography studies were performed concurrently with CMRI studies, on commercially available Vivid E9 machines (GE Healthcare, Norway). All measurements and analysis were performed offline, using an EchoPAC Clinical Workstation (GE Healthcare, Version 12). LVEF and LV volumes were measured by Simpson's biplane method and averaged over three consecutive cardiac cycles.

## Cardiac MRI acquisition protocol and analysis

Cardiac MRI was performed using standard multi-sequence protocol with image sequences obtained during breath hold using a commercially available MRI 1.5T scanner (Siemens Symphony, Germany), [26] as previously described [18]. Commercially available software (CMR 42, version 4.1.8, Circle Cardiovascular Imaging, Calgary, Canada) was utilised to quantify IS and MVO [2]. The mean signal threshold versus reference myocardium technique was used to evaluate scar characteristics with the hyper-enhanced area (defined as having a signal intensity threshold $\geq$ 5SD above region of interest of normal 'nulled' myocardium) defined as 'core' IS [26–28]. LV scar size was expressed as the percentage of infarct mass of total LV mass. Areas of hypo-enhancement with surrounding hyper-enhanced myocardium on LGE inversion recovery sequences were used to determine areas of MVO [4, 29]. Myocardial area at risk was calculated as the percentage of the hyper-enhanced myocardium volume (2SD of ROI) on T2 STIR sequences / the total myocardial volume. Myocardial Salvage Index (MSI) was subsequently calculated as: [myocardial area at risk–infarct core percentage] [18]. Measurements of IS, MVO, and T2 STIR sequences were independently performed by JO and TN, blinded to ECG analysis.

## Clinical follow-up

Late clinical outcomes were obtained via contact with cardiologists, general practitioners or direct contact for patients at 24 months. The primary clinical outcome was a hierarchical combination of all-cause mortality, stroke and new or worsening of heart failure, defined as a major adverse cardiovascular event (MACE) which has been previously described [2].

## Statistical analysis

Statistical analysis was conducted using Statistical Product and Service Solutions (SPSS, version 25, Chicago, Illinois). Patients were divided into two groups by infarct location. Presence of MVO on early MRI was used to divide patients for separate analysis. Categorial data is presented as count (percent) whilst continuous data is presented as median [inter-quartile range (IQR)]. Significant differences between groups were examined using Mann-Whitney U Test (for continuous data) and Pearson Chi-square or Fischer's Exact Test (as appropriate, for categorial data). A p value of <0.05 was considered statistically significant. Spearman's rank test was used to assess correlation between Selvester QRS Score and CMRI measured IS as well as between acute Selvester QRS score and 48-hour hsTnT levels. Agreement between measurements for Selvester QRS Score and CMRI measured IS was evaluated by Bland-Altman analysis. Patients were categorized into approximate tertiles according to their estimated IS by acute Selvester QRS Score, 48-hour hsTnT levels and acute CMRI IS. Kaplan-Meier curves were

subsequently constructed to compare combined MACE rates among these stratified subgroups and differences assessed using the log-rank test. Binary logistic regression analysis was performed to construct the final model using a backward selection with application of alpha cut off values of p<0.05 for initial models, to determine factors predictive of combined MACE. Early and follow-up CMRI IS, acute and follow-up Selvester QRS score as well as 48–72 hour hsTnT levels were included in the initial model.

## Results

### Patient population

Of 233 patients with STEMI undergoing CMRI and treated by PCI during initial hospitalization, 22 had prior MIs, and the remainder had initial MIs, 115 had anterior STEMIs and 96 non-anterior STEMIs (Fig 1). These 211 patients (183 males), with a median age 57 [50–64] years, had similar clinical characteristics with respect to MI location (Table 1). Primary PCI was performed on 165 patients (78%) whereas 46 (22%) had a pharmaco-invasive strategy. Patients with anterior MIs had more intra-aortic balloon pump use (12 vs. 1, p<0.01); see Table 2 for other angiographic and procedural characteristics.

### Infarct size assessment

Acute IS determined by Selvester QRS scores was 12% [6–21%] of the left ventricular (LV) myocardium for anterior and 3% [0–6%] for non-anterior STEMI, on 12-lead ECGs at 3.7 days (median; IQR [2.0–6.4]) post-STEMI. Follow-up IS at median 55 [46–66] days for 202 patients was 12% [6–18%] for anterior and 3% [0–6%] for non-anterior MI according to ECG criteria. Among 211 patients CMRI–determined IS at median 4 [2–7] days (early), was 11% [5.9–17%] for anterior and 7.4% [4.5–11%] for non-anterior STEMI. Follow-up IS by CMRI in 202 patients (median 54 [46–63] days), was 7.9% [4.4–13%] for anterior MIs and 5.9% [3.7–9.1%] for non-anterior MIs. Area at risk, MSI and IS measured by both Selvester QRS score and CMRI was significantly different between anterior and non-anterior MI (Table 3).

**Table 1. Baseline clinical characteristics.**

|  | All patients n = 211 | Location of Myocardial Infarction | | p value |
|---|---|---|---|---|
|  |  | Anterior MI n = 115 | Non-Anterior MI n = 96 |  |
| Age (years) | 57 [50–64] | 56 [50–64] | 55 [50–64] | 0.81 |
| Male gender | 183 (87) | 98 (85) | 85 (89) | 0.48 |
| Body mass index (kg/m$^2$) | 27 [25–30] | 27 [25–30] | 26 [24–30] | 0.62 |
| Hypertension | 95 (45) | 55 (48) | 40 (42) | 0.37 |
| Diabetes Mellitus | 39 (19) | 24 (21) | 15 (16) | 0.33 |
| Hyperlipidemia | 89 (42) | 50 (44) | 39 (41) | 0.68 |
| Cigarette Smoking (current or >10 pack year history) | 126 (60) | 65 (57) | 61 (64) | 0.30 |
| History of Angina | 33 (16) | 19 (17) | 14 (15) | 0.70 |
| Prior PCI | 7 (3.3) | 4 (3.5) | 3 (3.1) | 0.89 |
| Prior CVA or PVD | 9 (4.3) | 4 (3.5) | 5 (5.2) | 0.54 |
| Cardiac Arrest at Presentation | 7 (3.3) | 6 (5.2) | 1 (1) | 0.09 |
| Killip Class >1 | 12 (5.7) | 8 (7.0) | 4 (4.2) | 0.38 |

* Categorical variables are shown as count (%). Continuous variables are shown as median [inter-quartile range] CVA, Cerebrovascular Accident; PCI, Percutaneous Coronary Intervention; PVD, Peripheral Vascular Disease. # Coronary artery ≥ 70% stenosis

**Table 2. Angiographic and procedural characteristics.**

| | All patients n = 211 | Location of Myocardial Infarction | | p value |
| | | Anterior MI n = 115 | Non-Anterior MI n = 96 | |
| --- | --- | --- | --- | --- |
| Symptom to Reperfusion Time (mins) | 232 [143–414] | 244 [145–430] | 216 [142–356] | 0.47 |
| Culprit Artery | | | | |
| LAD | 115 (55) | 115 (100) | 0 (0) | - |
| LCx | 23 (11) | 0 (0) | 23 (24) | - |
| RCA | 73 (34) | 0 (0) | 73 (76) | - |
| Occluded Culprit Artery | 125 (59) | 59 (51) | 66 (69) | **0.01** |
| No. of diseased arteries# | | | | |
| Single Vessel | 138 (65) | 79 (69) | 59 (62) | 0.27 |
| Double Vessel | 59 (28) | 30 (26) | 29 (30) | 0.51 |
| Triple Vessel | 14 (7) | 6 (5.2) | 8 (8) | 0.37 |
| Reperfusion Strategy | | | | |
| Primary PCI | 165 (78) | 87 (76) | 78 (81) | 0.33 |
| Pharmacoinvasive | 46 (22) | 28 (24) | 18 (19) | 0.33 |
| Rescue | 12 (5.7) | 8 (7) | 4 (4) | 0.38 |
| Procedural GPIIb/IIIa use | 57 (27) | 27 (24) | 30 (31) | 0.21 |
| Bare Metal Stent Use | 111 (53) | 41 (36) | 70 (73) | **<0.01** |
| Drug Eluding Stent Use | 90 (43) | 69 (60) | 21 (22) | **<0.01** |
| Total Stented Length (mm) | 22 [18–29] | 22 [18–30] | 22 [15–28] | 0.29 |
| TIMI 3 flow post culprit PCI | 194 (92) | 105 (91) | 89 (93) | 0.71 |
| IABP Use | 13 (6.2) | 12 (10) | 1 (1.1) | **<0.01** |

* Categorical variables are shown as count (%). Continuous variables are shown as median [inter-quartile range] IABP, Intra-Aortic Balloon Pump; LAD, Left Anterior Descending Artery; LCx, Left Circumflex Artery; PCI, Percutaneous Coronary Intervention; RCA, Right Coronary Artery≥ 70% stenosis.

The correlation between CMRI and Selvester QRS score IS, among patients with anterior infarcts MI were r = 0.316, p<0.01 early, and at r = 0.320, p<0.01 at follow-up (Fig 2).Associations between QRS score and CMRI-determined IS for non-anterior MI were r = 0.236, p = 0.021 early, and r = 0.095, p = 0.373 at follow-up. Bland-Altman plots showed that IS tended to be overestimated by Selvester QRS scoring compared to CMRI at all-time points (supplementary Fig 1). It also showed moderate correlations between ECG and CMRI-determined IS–r = 0.419, p<0.01 early and r = 0.322, p<0.01 at follow-up. Echocardiography found LV ejection fraction improved from 48% [42–54] to 52% [44–60] over 2 months (Table 3).

Patients with MVO on CMRI had larger infarct sizes as measured by both Selvester QRS score and CMRI compared to patients without MVO at both time periods (p<0.01 for all comparisons). Additionally, area at risk was significantly higher in patients with MVO whilst MSI was significantly higher in patients without MVO as measured by both Selvester QRS score and CMRI (p<0.01 for all comparisons) (Table 4). Patients with MVO showed a trend towards later reperfusion times (p = 0.06).

On acute Selvester QRS scoring, 50 (24%) patients had a score of 0, 11 of which were anterior and 39 were non-anterior MI. The median CMRI measured IS of these patients was 4.6% [3.3–8.4%]. Moreover, 64 patients (30%) had ≥2-point differences in acute and follow-up QRS scores, which represents ≥6% of LV myocardium. The CMRI measured IS difference for these patients was only a median of 1.46% [0.2–3.65%].

Median hsTnT levels at 48–72-hour post-STEMI were 2658ng/L [1640–3982] for anterior and 1955ng/L [1077–3363] for non-anterior STEMI; p = 0.03 (Table 3). Correlation between

**Table 3. ECG, TTE, MRI and hsTnT parameters.**

| | All patients n = 211 | Location of Myocardial Infarction | | p value |
| --- | --- | --- | --- | --- |
| | | Anterior MI n = 115 | Non-Anterior MI n = 96 | |
| Maximum Lead ST-segment Recovery ≥50%† | 154 (73) | 78 (68) | 76 (79) | <0.01 |
| Maximum Lead ST-segment Recovery ≥70%† | 90 (43) | 37 (32) | 53 (55) | <0.01 |
| Total ST-segment deviation recovery ≥50%† | 150 (71) | 72 (63) | 78 (81) | <0.01 |
| Total ST-segment deviation recovery ≥70%† | 102 (48) | 42 (37) | 60 (63) | <0.01 |
| Selvester Area at Risk (%LV) | 21 [18–33] | 27 [21–36] | 21 [15–21] | <0.01 |
| Selvester Infarct Size (%LV) | | | | |
| Acute ECG | 6 [3–15] | 12 [6–21] | 3 [0–6] | <0.01 |
| Follow-up ECG* | 6 [0–15] | 12 [5–18] | 3 [0–6] | <0.01 |
| Acute Selvester MSI^ | 60 [43–94] | 48 [30–71] | 86 [64–100] | <0.01 |
| LVEF acute | 48 [42–54] | 46 [38–53] | 49 [45–55] | <0.01 |
| LVEF followup | 52 [44–60] | 49 [39–60] | 54 [47–60] | 0.03 |
| MRI T2 STIR Area at Risk (%LV)# | 41 [37–47] | 45 [41–49] | 38 [32–42] | <0.01 |
| MRI Infarct Size (%LV) | | | | |
| Acute MRI | 8.9 [5.1–14] | 11 [5.9–17] | 7.4 [4.5–11] | <0.01 |
| Follow-up MRI* | 6.9 [4.0–11] | 7.9 [4.4–13] | 5.9 [3.7–9.1] | 0.01 |
| Acute MRI MSI# | 77 [67–86] | 75 [65–86] | 78 [70–87] | 0.46 |
| Acute MRI LVEF | 47 [40–53] | 46 [37–52] | 49 [44–54] | <0.01 |
| Microvascular Obstruction | 97 (46) | 62 (54) | 35 (37) | 0.01 |
| 48 hour hsTnT (ng/L) | 2344 [1200–3561] | 2658 [1640–3982] | 1955 [1077–3363] | 0.03 |
| 72 hour hsTnT (ng/L) | 2307 [1220–3491] | 2513 [1277–3891] | 2017 [1188–3024] | 0.04 |

- Categorical variables are shown as count (%). Continuous variables are shown as median [inter-quartile range]ECG, Electrocardiogram; hsTnT, High-Sensitivity Troponin-T; MRI, Magnetic Resonance Imaging; MSI, Myocardial Salvage Index; STIR, Short Tau Inversion Recovery; LVEF, Left Ventricular Ejection Fraction; measured on Transthoracic Echocardiography

† 199 cases included (111 anterior and 88 non-anterior)

* 202 cases included (110 anterior and 92 non-anterior)'

^ 209 cases included (115 anterior and 94 non-anterior)'

# 195 cases included (106 anterior and 89 non-anterior)

acute Selvester QRS score and 48–72-hour hsTnT levels was r = 0.311, p = <0.01 (r = 0.265, p = <0.01 for anterior and r = 0.305, p<0.01 for non-anterior MIs) (Fig 2D).

## Late outcomes

At 24-month follow-up there were 5 deaths, 4 strokes and 15 readmissions for heart failure, occurring in 20 patients (9%). Two patients suffered sudden death, two died of congestive heart failure and one died due to malignancy. Of the 15 patients readmitted for heart failure, 9 had the highest tertile of QRS scores, 11 had the highest tertile of 48-hour hsTnT levels and 12 had the highest tertile of CMRI-measured IS. These patients had a mean LVEF on discharge of 37.5% compared to the mean of the whole cohort of 47.5%. Tertiles of acute Selvester QRS score, 48-hour hsTnT levels and acute CMRI were all predictive of composite MACE (all comparisons log-rank p<0.01) (Fig 3). Tertiles of acute Selvester QRS score, 48-hour hsTnT levels and acute CMRI were all predictive of readmission for heart failure (Table 5).

## Logistic regression analyses

Binary Logistic Regression analyses were undertaken to examine MACE at 24 months (Table 6). Multivariate analysis showed 72-hour hsTnT levels were most predictive of

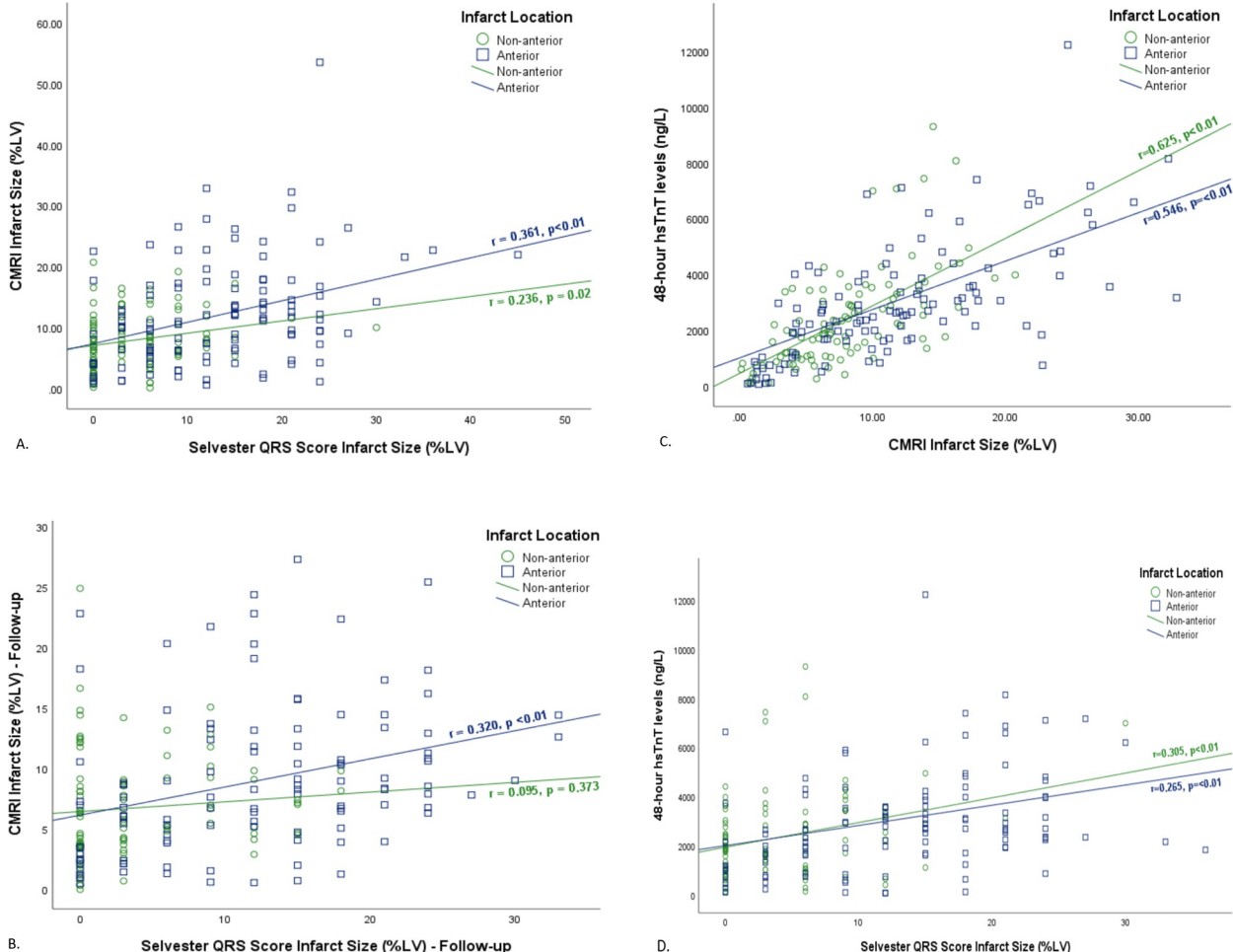

**Fig 2. Scatter plots of binary correlations between Selvester scores, Troponin T levels and CMRI derived infarct size.** A: Selvester QRS score and CMRI infarct size–Acute (n = 211). B: Selvester QRS score and CMRI infarct size–Follow-up (n = 199). C: Acute CMRI Infarct Size and 48-hour hsTnT levels (n = 209). D: Acute Selvester QRS score and 48-hour hsTnT levels (n = 209). Green Circles: non-anterior infarction patients. Blue Squares: anterior infarction patients. Green line: regression line for non-anterior MI. Blue line: regression line for anterior MI.

24-month MACE with every 1000 unit increase in 72-hour hsTnT levels increasing likelihood of MACE by 1.83.

## Discussion

Infarct size is an important determinant of late outcomes after an initial STEMI. In routine practice, CMRIs are often not readily available, are relatively expensive, and are precluded in certain patients including those with poor kidney function, are claustrophobic, or have body habitus preventing adequate scans. Our current study extends our earlier work demonstrating 48-72h hsTnT levels strongly correlating with CMRI measured IS [18]. We have now shown a moderate correlation between Selvester QRS score derived IS with that determined by CMRI. Selvester QRS scoring is a simple, readily available tool which, together with 48-72h hsTnT levels and echocardiography, could provide a cheap and readily assessable measures of IS post-STEMI, as ECGs are performed on all patients with STEMI on presentation and subsequently during initial hospitalisation. Selvester QRS score and CMRI taken prior to discharge as well

**Table 4. Cardiac MRI–microvascular obstruction.**

|  | MVO<br>n = 97 | No MVO<br>n = 114 | *p* value |
|---|---|---|---|
| Selvester Area at Risk (%LV) | 27 [21–39] | 21 [15–30] | <**0.01** |
| MRI T2 STIR Area at Risk (%LV)* | 44 [38–48] | 41 [34–45] | <**0.01** |
| Selvester Infarct Size (%LV) |  |  |  |
| Acute ECG | 12 [3.0–18] | 6.0 [0–9.0] | <**0.01** |
| Follow-up ECG# | 12 [3.0–18] | 3.0 [0–9.0] | <**0.01** |
| MRI Infarct Size (%LV) |  |  |  |
| Acute MRI | 13 [9.2–18] | 6.3 [3.4–8.9] | <**0.01** |
| Follow-up MRI# | 9.2 [6.7–14] | 4.7 [2.4–7.6] | <**0.01** |
| Acute Selvester MSI | 50 [38–83] | 77 [50–100] | <**0.01** |
| Acute MRI MSI | 70 [61–77] | 84 [75–91] | <**0.01** |
| Time to reperfusion (mins) | 250 [156–441] | 195 [134–360] | 0.06 |

ECG, Electrocardiogram; LV, Left Ventricular, MRI, Magnetic Resonance Imaging; MSI, Myocardial Salvage Index;
STIR, Short Tau Inversion Recovery

\* 195 cases included (87 MVO and 108 No MVO)'

# 202 cases included (94 MVO and 108 No MVO)'

† 186 cases included (84 MVO and 102 No MVO)'

‡ 187 cases included (84 MVO and 103 No MVO)

as 48–72-hour hsTnT levels were all predictive of 24-month MACE with 72-hour hsTnT levels being most predictive on multivariate binary logistic regression.

In our study, Selvester QRS scoring and CMRI IS measurements correlated moderately well both early, and at 2-months however correlation for non-anterior STEMIs was weak. Prior studies have reported correlation co-efficients ranging from 0.39–0.79 in the acute period, [11, 13, 15–17, 30] and between 0.43–0.78 at follow up [12, 14–17, 30]. Bang et al. [30] found similar results, with weak correlations between Selvester QRS score and CMRI IS at 1–2 days (r = 0.24, p = 0.32), 1 (r = 0.13, p = 0.6) and 6 months (r = 0.15, p = 0.54) post-STEMI. However, Engblom et al. [14, 15], who also correlated CMRI and Selvester QRS score estimated IS in the acute period, and found a strong correlation (r = 0.72, p = 0.004). However, both studies had very few non-anterior STEMIs—20 inferior STEMI out of a total of 31 in Bang et al. and 14 inferior STEMIs out of a total of 25 (all RCA occlusions) in Engblom et al.

Microvascular obstruction, frequently associated with the no-reflow phenomenon, has been found to be independently associated with a poorer prognosis [5]. We found that patients with MVO have significantly larger MIs, as measured by Selvester QRS scoring and CMRI, and patients with anterior MIs more often MVO. This is consistent with previous studies and suggests these findings may be merely a reflection of the extent of IS [16, 31]. We also found that patients with MVO have higher area at risk and lower MSI compared to those without. This suggests MVO is more likely to develop in patients with occlusions supplying a larger myocardial territory. Of note, 39/80 patients (49%) with a ≥2-point difference between acute and follow-up Selvester QRS score had MVO. MVO may alter electrical properties, cause conduction changes and impact scar remodelling which could confound QRS scoring at both acute and follow-up ECG assessments. The impact of MVO presence on change between acute and follow-up Selvester QRS scores may warrant further exploration.

Tjandrawidjaja et al. found in a cohort of more than 4,000 patients that IS estimated by Selvester QRS score at discharge was an independent predictor of adverse clinical outcomes at 90-days [32]. Several other studies have assessed prognostic utility of Selvester QRS scoring

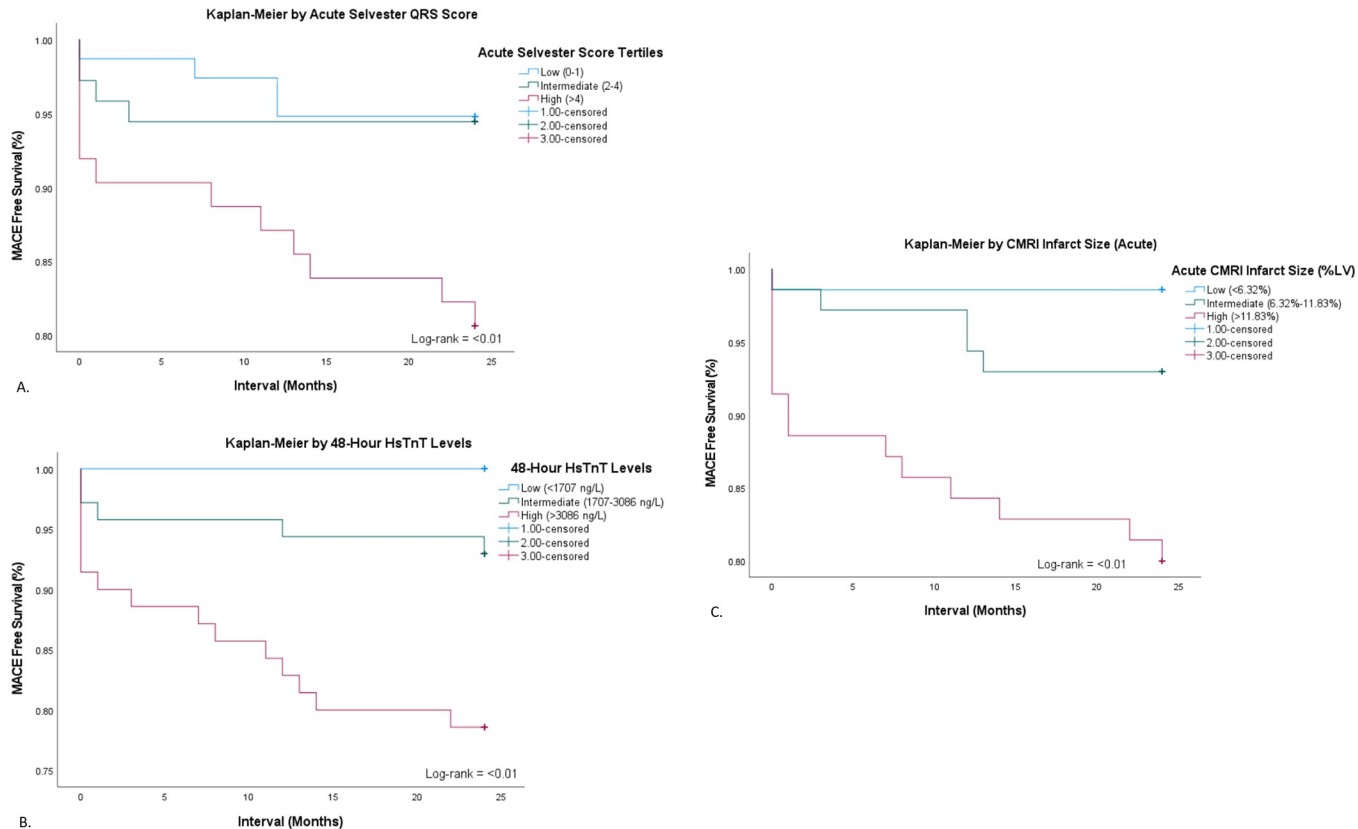

**Fig 3. Kaplan-Meier curves of freedom from MACE.** A: Comparisons are shown between tertiles of: Acute QRS Selvester scores. B: 48-Hour hsTnT levels. C: Acute CMRI infarct size (Log-rank p = 0.173). Blue Line (low), Green Line (intermediate) and Red Line (high) tertiles.

post-STEMI [32–38], though inter-study comparisons are confounded by QRS scoring being performed at various time points (admission or discharge), different clinical outcome times being utilised and lack of comparisons with CMRI or another standard IS measure. We have utilised the modified Selvester QRS which has enabled inclusion of patients with ECG conduction abnormalities such as left and right bundle branch block, left anterior fascicular block and left ventricular hypertrophy. We also included patients with multi-vessel disease and treated by both primary and pharmaco-invasive PCI strategies. While echocardiography was useful in showing changes in ejection fractions, it was not as precise in determining IS.

**Table 5. 24-Month clinical outcomes according to tertiles of acute Selvester QRS score, 48-hour hsTnT levels and cardiac MRI infarct size.**

|  | Acute Selvester QRS Score n = 211 | | | | 48-hour hsTnT Levels (ng/L) n = 211 | | | | Cardiac MRI Infarct Size (%LV) n = 211 | | | |
|---|---|---|---|---|---|---|---|---|---|---|---|---|
|  | **0–1** n = 77 | **2–4** n = 72 | **>4** n = 62 | **p value** | **<1707** n = 70 | **1707–3086** n = 71 | **>3086** n = 70 | **p value** | **<6.32** n = 70 | **6.32–11.8** n = 71 | **>11.8** n = 70 | **p value** |
| All Cause Death | 1 | 0 | 4 | **0.04** | 0 | 1 | 4 | 0.07 | 0 | 2 | 3 | 0.24 |
| CVA | 0 | 2 | 2 | 0.31 | 0 | 0 | 4 | **0.02** | 1 | 1 | 2 | 0.77 |
| Readmission for Heart Failure | 4 | 2 | 9 | **0.02** | 0 | 4 | 11 | **<0.01** | 0 | 3 | 12 | **<0.01** |
| Composite MACE | 4 | 4 | 12 | **<0.01** | 0 | 5 | 15 | **<0.01** | 1 | 5 | 14 | **<0.01** |

CVA, Cerebrovascular Accident; hsTnT, High-Sensitivity Troponin-T; LV, Left Ventricular; MACE, Major Adverse Cardiovascular Outcomes; MRI, Magnetic Resonance Imaging.

**Table 6. Binary logistic regression predictors of 24-month composite MACE.**

| | *Univariate* | | | *Multivariable* | | |
|---|---|---|---|---|---|---|
| | **OR** | **95% CI** | **P** | **OR** | **95% CI** | **P** |
| 48-Hour hsTnT levels* | 1.44 | 1.18, 1.76 | **<0.01** | | | |
| 72-Hour hsTnT levels* | 1.83 | 1.44, 2.32 | **<0.01** | 1.91 | 1.22, 2.97 | **<0.01** |
| Acute Selvester QRS Score | 1.28 | 1.10, 1.49 | **<0.01** | | | |
| Follow-up Selvester QRS Score | 1.24 | 1.05, 1.47 | **0.01** | | | |
| Acute MRI Infarct Size | 1.16 | 1.08, 1.23 | **<0.01** | | | |
| Follow-up MRI Infarct Size | 1.22 | 1.12, 1.32 | **<0.01** | | | |
| Acute TTE LVEF | 0.87 | 0.82, 0.93 | **<0.01** | | | |
| Follow-up TTE LVEF | 0.94 | 0.91, 0.97 | **<0.01** | | | |
| Total ST-segment deviation recovery ≥70% | 0.48 | 0.18, 1.25 | 0.13 | | | |
| Final TIMI Flow = 3 | 1.74 | 0.22, 13.8 | 0.61 | | | |

* for every 1000-unit increase

hsTnT, high-Sensitivity Troponin-T; LVEF, Left Ventricular Ejection Fraction; MACE, Major Adverse Cardiovascular Outcomes; MRI, Magnetic Resonance Imaging; TTE, Transthoracic Echocardiography

Our study, to our knowledge, is the first to report late clinical outcomes for a cohort of patients having 3 modalities of IS assessment, Selvester QRS scores, hsTnT levels and CMRI. We have found a positive correlation between tertiles of acute Selvester QRS score, 48-hour hsTnT levels and acute CMRI IS and 24-month MACE free survial (Fig 3). The highest tertiles of all correlated with readmission for heart failure (Table 5). These findings allowed analysis of prognostic associations of all parameters and MACE via logistic regression. Whilst all predicted 24-month composite MACE on univariate analysis, 72-hour hsTnT levels were found to be the most predictive on multivariate analysis (Table 6). Our findings should be interpreted with caution given the relatively small numbers of MACE with 24 events in 20 patients.

Our study has other limitations inherent in STEMI patients undergoing CMRI, including those at highest risk not being able to undergo early CMRI and were thus not included. This probably led to our median patient age of 57 years which is younger than many contemporary STEMI cohorts. While ours is the largest study in the era of reperfusion predominantly by primary PCI to compare IS measured by Selvester QRS scores, hsTnT levels and CMRI in anterior and non-anterior STEMI patients at both early and follow-up time-points, we still did not have sufficient study power to adequately evaluate these assessments of IS, particularly for non-anterior MIs, as predictors of late outcomes. After admission for STEMI, good clinical practice for a few decades in coronary care units (CCU) has been to measure cardiac biomarkers ~3 times in the first 24 hours and daily for at least 48 hours. The performance and/or timing of hsTnT (or I) testing is not mandatory. Thus, our study protocol reflects collection of hsTnT at particular time intervals in our CCU, which may have significantly influenced our correlations. While echocardiography is commonly used to assess LVEF after STEMI it did not achieve significance in multi variable models.

## Conclusions

Simple measures of infarct size by Selvester QRS scores and hsTnT levels post-STEMI correlated well with CMRI-determined IS. Additionally, all parameters predicted MACE at 24-months. Using routinely performed, widely available and inexpensive indices of infarct size assessment offers a potential simple avenue for multi-modal risk stratification post-STEMI, leaving the more expensive CMRI-based IS assessment for selective use in those at higher risk.

## Supporting information

**S1 Fig.**
(ZIP)

**S1 Data.**
(XLSX)

## Author Contributions

**Conceptualization:** Lokesh Sharma, Amir Faour, Tuan Nguyen, Hany Dimitri, James Otton, Sonya Burgess, Craig Juergens, John French.

**Formal analysis:** John French.

**Investigation:** Lokesh Sharma, Tuan Nguyen, Giau Vo, Sonya Burgess, John French.

**Methodology:** Lokesh Sharma, Amir Faour, Tuan Nguyen, Hany Dimitri, James Otton, Sonya Burgess, Craig Juergens, John French.

**Project administration:** Tuan Nguyen, Hany Dimitri, John French.

**Resources:** Tuan Nguyen.

**Supervision:** Amir Faour, Tuan Nguyen, Hany Dimitri, James Otton, Sonya Burgess, Craig Juergens, John French.

**Validation:** Amir Faour, Tuan Nguyen, James Otton, Sonya Burgess, Craig Juergens, John French.

**Writing – original draft:** Lokesh Sharma, Tuan Nguyen, John French.

**Writing – review & editing:** Lokesh Sharma, Amir Faour, Tuan Nguyen, Hany Dimitri, Giau Vo, James Otton, Sonya Burgess, Craig Juergens, John French.

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
