## [Decision Letter · Decision Letter 0]

11 Jun 2024

PONE-D-24-11346Simple Indices of Infarct Size post ST-Elevation Myocardial Infarction (STEMI) provides similar Risk Stratification to Cardiac MRIPLOS ONE

Dear Dr. French,

Thank you for submitting your manuscript to PLOS ONE. After careful consideration, we feel that it has merit but does not fully meet PLOS ONE’s publication criteria as it currently stands. Therefore, we invite you to submit a revised version of the manuscript that addresses the points raised during the review process.

We look forward to receiving your revised manuscript.

Kind regards,

Martin E. Matsumura, MD

Academic Editor

PLOS ONE

"N/A"

3. In the online submission form, you indicated that Deidentified data may be obtained from the authors on request.

Additional Editor Comments:

Your work has been deemed to have significant merit by reviewers. Please respond to all reviewer comments, thank you. Furthermore, the editor has some specific requests:

1. Please expand in the methodology the definition of pharmaco-intervention. I assume this means planned thrombolytic treatment followed by transfer to a PCI center.  However, this term and the specific determination of when it is used (ie, US recs specify consideration of D2B is anticipated to be >120min).  

2. I would appreciate if you had a bit more regarding the methodology of the Selvester score in the methods- perhaps a few lines regarding how it is calculated/what specific parameters are examined.  The original reference may not be readily available and many/most readers may not be familiar with the score.  

Reviewers' comments:

Reviewer's Responses to Questions

**Comments to the Author**

1. Is the manuscript technically sound, and do the data support the conclusions?

Reviewer #1: Partly

2. Has the statistical analysis been performed appropriately and rigorously? 

Reviewer #1: Yes

3. Have the authors made all data underlying the findings in their manuscript fully available?

Reviewer #1: Yes

4. Is the manuscript presented in an intelligible fashion and written in standard English?

Reviewer #1: Yes

5. Review Comments to the Author

**Reviewer #1: **Thank you for the opportunity to review the paper titled “Simple indices of infarct size post ST-Elevation Myocardial Infarction (STEMI) provides similar risk stratification to Cardiac MRI”

I want to commend the authors for undertaking research that is pragmatic and likely to benefit contemporary clinical care in many centres, rather than just large tertiary or quintenary centres. I believe this data deserves publication.

General comments:

- I agree with the authors than cardiac MRI is the gold standard for assessing infarct size and myocardial function post myocardial infarction. It also adds invaluable insight for each patient by allowing for assessment of microvascular obstruction as well as myocardium at risk. I do believe this helps us understand why some patients are likely to recover more myocardial function than others. These features are not always apparent at angiography or on echocardiography.

- I commend the authors for utilizing ECG parameters as an estimate of infarct size as this is a readily available test for all centres who are able to treat myocardial infarctions.

- The sample size must also be recognised as excellent given the requirement for participants undergoing both and acute and delayed cardiac MRI in this prospective cohort.

- I would encourage the authors to ensure that the structure of the article flows. There are occasions where the style changes from one sentence to the next making it slightly harder to read.

- I would encourage the authors to try to minimize the colloquial terminology such as “infarcts”. Instead I would recommend replacing this with either myocardial infarction or MI (given you have used this abbreviation throughout the text already).

Specific comments on text:

- Section “statistical analysis”

o Line 136; need a comma after “as appropriate”

- Section “patient population”,

o Lines 154-155 needs rewording e.g. These 211 patients (183 male), with a median age of 57 [50-64] years, had similar clinical characteristics with respect to infarct location (table 1)

- Section “infarct size assessment”

o Line 163; add “for” in front of non-anterior infarcts e.g. “and 3% [0-6%] for non-anterior infarcts according to ECG criteria”

o Line 166; reword “(median 54 [46-63] days late)” – suggest removing the word late

o Lines 170-178; suggest rewording paragraph – you are presenting similar data for this section however each sentence has a different style / presentation – all needs to be consistent i.e. in one sentence the r values are in brackets, in others it is in the main text.

o Lines 179-183; Was there any correlation between the time to perfusion and MVO or the type of reperfusion and MVO

o Line 184; I would reword this to make clearer e.g.

On acute Selvester QRS scoring 50 patients (24%) had a score of 0, 11 of which were anterior and 39 were non-anterior MI

o Line 187-188; need to reword

The CMRI measured infarct size difference for these patients was only a median of 1.46%

o Line 192; check MIs spelling

- Section Late outcomes;

o Line 195; could remove word “witnessed”

o You have stated that of the 15 with hospitalisation for heart failure, pre-discharge echocardiogram demonstrated an LVEF of <45% in12/15 (which is more that either QRS or hsTnT) 0 should this not be included in the analysis of MACE and readmission for heart failure? Or have you excluded it because it was not part of your primary analysis nor did it perform well in univariate analysis?

Discussion / conclusion

- I think you need to be careful stating that QRS derived infarct size correlates well with CMR derived infarct size.

o Your previous work definitely demonstrated a strong correlation between hsTnT and CMR derived infarct size however, although there is a correlation in your current study for QRS and CMR IS, I struggle to appreciate the same level of clinically significant correlation that would allow useful predictions.

o The r value for the current correlation was only 0.316 and 0.320 early and at follow up respectively in anterior infarcts. The figures demonstrate quite significant heterogeneity and I would actually hazard there is very poor correlation in anterior infarcts and even worse in non-anterior infarcts

- I agree that, although each readily available test (hsTnT, ECG and Echo) are useful, it may take a prediction tool / score that incorporates all of these factors into providing the same information as a cardiac MRI with regards to clinical prediction

- Paragraph lines 221 to 229; needs rewording / clarification

o Line 221; “in this study Selvester QRS scoring and CMR IS measurements correlated… well? Poorly?”

o Line 224; “Correlation with non-anterior STEMIs was weak” this statement needs expanding or referencing – e.g. in the current study…

o Line 227; ensure continuity with wording – you have used “Selvester QRS score” throughout the rest of the paper, you need to either change all to just “Selvester score” after the initial “Selvester QRS score” or add QRS here

o Line 226-227; I would also reword this sentance e.g.

However, Engblom et al who also correlated CMRI and Selvester QRS score estimated IS in the acute period, and also found a strong correlation (r=0.72, p=0.004).

o Line 228-229; this sentence is a bit hard to read and rewording may be useful e.g.

Both studies had very few non-anterior STEMIs in their cohort

You haven’t told us how many were in the cohort overall so 20 and 14 doesn’t mean a lot i.e. is it 20 out of 50 or 20 out of 500 – I don’t think you need to specify this, just note that there were few non-anterior STEMIs

- Line 250; the sentence “As we also included patients with multi-vessel disease and treated with both primary and pharmaco-invasive PCI strategies” – this sentence doesn’t appear to fit on its own. I appreciate you are trying to say your cohort was inclusive – you could just remove the word “As” at the start and it would make more sense.

Study limitations should include:

- As hsTnT is not a mandatory test performed for the investigation or management of STEMI, rigorous study protocol mandating collection of hsTnT at specific time points, as in this study, influences correlations significantly. When troponin is clinically acquired, the correlation with left ventricular dysfunction is significantly poorer.

o Consider reading: https://doi.org/10.1016/j.hlc.2022.07.014

Final Comments;

- I believe this study has merit. It helps to add further information to the assessment of long term outcomes in STEMI patients

- The authors have tried to cover a lot of analyses in this paper with

o Correlation of QRS score to CMR IS

o Correlation of hsTnT to CMR IS (extension of their previous work)

o Correlation of IS to late outcomes

- I would be cautious using a test to estimate an outcome (in this case IS) and then using that outcome to correlate with events (MACE)

o You have clearly demonstrated again that rigorously acquired hsTnT at defined time points correlates modestly with initial CMR IS (r=0.625, Figure 2C) in non-anterior STEMI and anterior STEMI (r=0.546) – somewhat surprising that non-anterior had a stronger correlation than anterior

o You have demonstrated that Selvester QRS score correlates fairly poorly with initial CMR and follow up CMR (r=0.361 and r=0.320 respectively) in anterior STEMI, with poorer correlation in non-anterior STEMI. hsTnT as correlates poorly with initial Selvester QRS score in both non-anterior STEMI (r=0.305) and anterior STEMI (r=0.265)

- You have clearly demonstrated however that using the tertiles of hsTnT, Selvester QRS score and CMR IS a fairly clear demarcation of risk.

o Unfortunately, as you have commented on in your discussion, the sample size is not adequate to make clear conclusions in this respect given the low number of MACE events which isn’t unexpected given the low level of LV dysfunction seen in this cohort.

6. PLOS authors have the option to publish the peer review history of their article (what does this mean?). If published, this will include your full peer review and any attached files.

Reviewer #1: **Yes: **Peter James McLeod

---

## [Author Response · Author response to Decision Letter 0]

29 Aug 2024

Please note we have submitted a document called "Responses to reviewer & editor comments" it contains the below, along with our revisions.

Responses to reviewer & editor comments:

1. Please expand in the methodology the definition of pharmaco-intervention. I assume this means planned thrombolytic treatment followed by transfer to a PCI center. However, this term and the specific determination of when it is used (ie, US recs specify consideration of D2B is anticipated to be >120min). 

We agree with the reviewer and have added the following to the methods:

“Pharmaco-invasive strategy was defined as fibrinolytic administration followed by rescue PCI or by scheduled angiography and PCI.”

2. I would appreciate if you had a bit more regarding the methodology of the Selvester score in the methods- perhaps a few lines regarding how it is calculated/what specific parameters are examined. The original reference may not be readily available and many/most readers may not be familiar with the score. 

We agree with the reviewer and have added the following to the methods:

“In brief, the Selvester QRS score is a 33-point score that estimates IS based on 38 parameters of the QRS waveform such as amplitude, duration and ratios. Each point correlates to 3% of infarcted LV myocardium”

Comments to the Authors:

1. Is the manuscript technically sound, and do the data support the conclusions?

Reviewer #1: Partly

We have amended the conclusions as per the reviewers’ comments below.

“Post-PCI treatment of STEMI, hsTnT levels measured ≥48h and Selvester QRS scoring correlated with CMRI-determined IS. These parameters predicted MACE at 24 months and should be routinely assessed for post-STEMI risk stratification.”

2. Has the statistical analysis been performed appropriately and rigorously?

Reviewer #1: Yes

3. Have the authors made all data underlying the findings in their manuscript fully available?

Reviewer #1: Yes

4. Is the manuscript presented in an intelligible fashion and written in standard English?

Reviewer #1: Yes

5. Review Comments to the Author

Reviewer #1: Thank you for the opportunity to review the paper titled “Simple indices of infarct size post ST-Elevation Myocardial Infarction (STEMI) provides similar risk stratification to Cardiac MRI”

I want to commend the authors for undertaking research that is pragmatic and likely to benefit contemporary clinical care in many centres, rather than just large tertiary or quintenary centres. I believe this data deserves publication.

General comments:

- I agree with the authors that cardiac MRI is the gold standard for assessing infarct size and myocardial function post myocardial infarction. It also adds invaluable insight for each patient by allowing for assessment of microvascular obstruction as well as myocardium at risk. I do believe this helps us understand why some patients are likely to recover more myocardial function than others. These features are not always apparent at angiography or on echocardiography.

- I commend the authors for utilizing ECG parameters as an estimate of infarct size as this is a readily available test for all centres who are able to treat myocardial infarctions.

- The sample size must also be recognised as excellent given the requirement for participants undergoing both and acute and delayed cardiac MRI in this prospective cohort.

- I would encourage the authors to ensure that the structure of the article flows. There are occasions where the style changes from one sentence to the next making it slightly harder to read.

We agree with the reviewer and have made small revisions to the flow of the manuscript.

- I would encourage the authors to try to minimize the colloquial terminology such as “infarcts”. Instead I would recommend replacing this with either myocardial infarction or MI (given you have used this abbreviation throughout the text already).

We agree with the reviewer and have changed all mention of “infarcts” to MI.

Specific comments on text:

- Section “statistical analysis”

o Line 136; need a comma after “as appropriate”

We have added a comma

- Section “patient population”,

o Lines 154-155 needs rewording e.g. These 211 patients (183 male), with a median age of 57 [50-64] years, had similar clinical characteristics with respect to MI location (table 1)

We have reworded the sentence as above.

- Section “infarct size assessment”

o Line 163; add “for” in front of non-anterior infarcts e.g. “and 3% [0-6%] for non-anterior infarcts according to ECG criteria”

We have reworded the sentence as above.

o Line 166; reword “(median 54 [46-63] days late)” – suggest removing the word late

We have removed the word “late” as suggested.

o Lines 170-178; suggest rewording paragraph – you are presenting similar data for this section however each sentence has a different style / presentation – all needs to be consistent i.e. in one sentence the r values are in brackets, in others it is in the main text.

We have edited the paragraph (see below) so the sentence structures are consistent:

“The correlation between CMRI and Selvester QRS score IS, among patients with anterior infarcts MI were r=0.316, p<0.01 early, and at r=0.320, p<0.01 at follow-up (Figure 2).Associations between QRS score and CMRI-determined IS for non-anterior MI were r=0.236, p=0.021 early, and r=0.095, p=0.373 at follow-up. Bland-Altman plots showed that IS tended to be overestimated by Selvester QRS scoring compared to CMRI at all-time points (supplementary Fig 1). It also showed moderate correlations between ECG and CMRI-determined IS – r=0.419, p<0.01 early and r=0.322, p<0.01 at follow-up.”

o Lines 179-183; Was there any correlation between the time to perfusion and MVO or the type of reperfusion and MVO

We thank the reviewer for their comments, and have added a row to Table 4 with a comparison p=0.06, of time to perfusion and MVO, which we suspect reflects our study’s power. We have added the following line to the results section:

“Patients with MVO showed a trend towards later reperfusion times (p=0.06)”

We have also run the statistics (Chi-square test) for Primary PCI and other reperfusion strategies vs presence of MVO as below. This was not statistically significant (Pearson Chi-square = 0.197), though this has not been included in the manuscript.

o Line 184; I would reword this to make clearer e.g.

On acute Selvester QRS scoring 50 patients (24%) had a score of 0, 11 of which were anterior and 39 were non-anterior Mis.

We have reworded the sentence as above.

o Line 187-188; need to reword

The CMRI measured infarct size difference for these patients was only a median of 1.46%

We have reworded the sentence as above.

o Line 192; check MIs spelling

The spelling has been corrected.

- Section Late outcomes;

o Line 195; could remove word “witnessed”

We have removed the word “witnessed”.

o You have stated that of the 15 with hospitalisation for heart failure, pre-discharge echocardiogram demonstrated an LVEF of <45% in12/15 (which is more that either QRS or hsTnT) 0 should this not be included in the analysis of MACE and readmission for heart failure? Or have you excluded it because it was not part of your primary analysis nor did it perform well in univariate analysis?

Our univariate analysis (Table 5) uses continuous LVEF % (not a dichotomous grouping of LVEF) as a predictor of MACE. In order to maintain consistency we have edited the results section to using mean LVEF as follows:

“Of the 15 patients readmitted for heart failure, 9 had the highest tertile of QRS scores, 11 had the highest tertile of 48-hour hsTnT levels and 12 had the highest tertile of CMRI-measured IS. These patients had a mean LVEF on discharge of 37.5% compared to the mean of the whole cohort of 47.5%.”

Discussion / conclusion

- I think you need to be careful stating that QRS derived infarct size correlates well with CMR derived infarct size.

o Your previous work definitely demonstrated a strong correlation between hsTnT and CMR derived infarct size however, although there is a correlation in your current study for QRS and CMR IS, I struggle to appreciate the same level of clinically significant correlation that would allow useful predictions.

o The r value for the current correlation was only 0.316 and 0.320 early and at follow up respectively in anterior infarcts. The figures demonstrate quite significant heterogeneity and I would actually hazard there is very poor correlation in anterior infarcts and even worse in non-anterior infarcts

We agree with the reviewer and have amended the discussion section as follows:

“Our current study extends our earlier work demonstrating 48-72h hsTnT levels strongly correlating with CMRI measured IS (18). We have now shown a moderate correlation between Selvester QRS score derived IS with that determined by CMRI.”

- I agree that, although each readily available test (hsTnT, ECG and Echo) are useful, it may take a prediction tool / score that incorporates all of these factors into providing the same information as a cardiac MRI with regards to clinical prediction

- Paragraph lines 221 to 229; needs rewording / clarification

o Line 221; “in this study Selvester QRS scoring and CMR IS measurements correlated… well? Poorly?”

o Line 224; “Correlation with non-anterior STEMIs was weak” this statement needs expanding or referencing – e.g. in the current study…

We have amended the sentence as follows:

“In our study, Selvester QRS scoring and CMRI IS measurements correlated moderately well both early, and at 2-months however correlation for non-anterior STEMIs was weak.”

o Line 227; ensure continuity with wording – you have used “Selvester QRS score” throughout the rest of the paper, you need to either change all to just “Selvester score” after the initial “Selvester QRS score” or add QRS here

We have added in “QRS” as suggested.

o Line 226-227; I would also reword this sentance e.g.

However, Engblom et al who also correlated CMRI and Selvester QRS score estimated IS in the acute period, and also found a strong correlation (r=0.72, p=0.004).

We have amended the sentence as above.

o Line 228-229; this sentence is a bit hard to read and rewording may be useful e.g.

Both studies had very few non-anterior STEMIs…

We have amended the sentence as above.

You haven’t told us how many were in the cohort overall so 20 and 14 doesn’t mean a lot i.e. is it 20 out of 50 or 20 out of 500 – I don’t think you need to specify this, just note that there were few non-anterior STEMIs

This has been amended as follows:

“Both studies had small non-anterior STEMIs - 20 inferior STEMI out of a total of 31 in Bang et al. and 14 inferior STEMIs out of a total of 25 (all RCA occlusions) in Engblom et al.”

- Line 250; the sentence “As we also included patients with multi-vessel disease and treated with both primary and pharmaco-invasive PCI strategies” – this sentence doesn’t appear to fit on its own. I appreciate you are trying to say your cohort was inclusive – you could just remove the word “As” at the start and it would make more sense.

We have amended the sentence as suggested.

Study limitations should include:

- As hsTnT is not a mandatory test performed for the investigation or management of STEMI, rigorous study protocol mandating collection of hsTnT at specific time points, as in this study, influences correlations significantly. When troponin is clinically acquired, the correlation with left ventricular dysfunction is significantly poorer.

o Consider reading: https://doi.org/10.1016/j.hlc.2022.07.014

We thank the reviewer for the reference and have revised the suggested edits to the above sentence in the limitations as follows: 

“After admission for STEMI, good clinical practice for a few decades in coronary care units (CCU) has been to measure cardiac biomarkers ~3 times in the first 24 hours and daily for at least 48 hours. The performance and/or timing of hsTnT (or I) testing is not mandatory. Thus, our study protocol reflects collection of hsTnT at particular time intervals in our CCU, which may have significantly influenced our correlations.”

Final Comments;

- I believe this study has merit. It helps to add further information to the assessment of long term outcomes in STEMI patients

- The authors have tried to cover a lot of analyses in this paper with

o Correlation of QRS score to CMR IS

o Correlation of hsTnT to CMR IS (extension of their previous work)

o Correlation of IS to late outcomes

- I would be cautious using a test to estimate an outcome (in this case IS) and then using that outcome to correlate with events (MACE)

o You have clearly demonstrated again that rigorously acquired hsTnT at defined time points correlates modestly with initial CMR IS (r=0.625, Figure 2C) in non-anterior STEMI and anterior STEMI (r=0.546) – somewhat surprising that non-anterior had a stronger correlation than anterior

o You have demonstrated that Selvester QRS score correlates fairly poorly with initial CMR and follow up CMR (r=0.361 and r=0.320 respectively) in anterior STEMI, with poorer correlation in non-anterior STEMI. hsTnT as correlates poorly with initial Selvester QRS score in both non-anterior STEMI (r=0.305) and anterior STEMI (r=0.265)

- You have clearly demonstrated however that using the tertiles of hsTnT, Selvester QRS score and CMR IS a fairly clear demarcation of risk.

o Unfortunately, as you have commented on in your discussion, the sample size is not adequate to make clear conclusions in this respect given the low number of MACE events which isn’t unexpected given the low level of LV dysfunction seen in this cohort.

---

## [Editor Report · Decision Letter 1]

16 Sep 2024

Simple Indices of Infarct Size post ST-Elevation Myocardial Infarction (STEMI) provides similar Risk Stratification to Cardiac MRI

PONE-D-24-11346R1

Dear Dr. French,

We’re pleased to inform you that your manuscript has been judged scientifically suitable for publication and will be formally accepted for publication once it meets all outstanding technical requirements.

Kind regards,

Martin E. Matsumura, MD

Academic Editor

PLOS ONE

Additional Editor Comments (optional):

Thank you for the detailed response to all editor and reviewer comments, your responses are adequate for acceptance

---

## [Editor Report · Acceptance letter]

8 Oct 2024

PONE-D-24-11346R1 

PLOS ONE

Dear Dr. French, 

I'm pleased to inform you that your manuscript has been deemed suitable for publication in PLOS ONE. Congratulations! Your manuscript is now being handed over to our production team.

Kind regards, 

on behalf of

Dr. Martin E. Matsumura 

Academic Editor

PLOS ONE